# Blockade of β2-Adrenergic Receptor Reduces Inflammation and Oxidative Stress in Clear Cell Renal Cell Carcinoma

**DOI:** 10.3390/ijms23031325

**Published:** 2022-01-25

**Authors:** Virginia Albiñana, Lucía Recio-Poveda, Pilar González-Peramato, Luis Martinez-Piñeiro, Luisa María Botella, Angel M. Cuesta

**Affiliations:** 1Centro de Investigaciones Biológicas Margaritas Salas, 28040 Madrid, Spain; vir_albi_di@yahoo.es (V.A.); luciarecio@hotmail.com (L.R.-P.); 2Centro de Investigación Biomédica en Red de Enfermedades Raras (CIBERER), ISCIII (Instituto de Salud Carlos III), 28029 Madrid, Spain; 3Department of Pathology, La Paz University Hospital, Autonomous University of Madrid, 28029 Madrid, Spain; mpilar.gonzalezperamato@salud.madrid.org; 4Department of Urology, Instituto de Investigación IdiPaz, 28029 Madrid, Spain; luis.mpineiro@salud.madrid.org; 5Department of Biochemistry and Molecular Biology, Faculty of Pharmacy, Complutense University of Madrid, 28040 Madrid, Spain

**Keywords:** von Hippel-Lindau, hemangioblastoma, ccRCC, β-adrenergic receptor, β-blocker, propranolol, ICI-118,551, inflammatory cytokines, reactive oxygen species (ROS), cancer therapy

## Abstract

Von Hippel-Lindau (VHL) syndrome is a rare inherited cancer disease where the lack of VHL protein triggers the development of multisystemic tumors such us retinal hemangioblastomas (HBs), CNS-HBs, and clear cell renal cell carcinoma (ccRCC). Since standard therapies in VHL have shown limited response, leaving surgery as the only possible treatment, targeting of the β2-adrenergic receptor (ADRB2) has shown therapeutic antitumor benefits on VHL-retinal HBs (clinical trial), VHL-CNS HBs, and VHL-ccRCC (in vitro and in vivo). In the present study, we wanted to look deep into the effects of the ADRB2 blockers propranolol and ICI-118,551 on two main aspects of cancer progression: (i) the changes on the inflammatory response of ccRCC cells; and (ii) the modulation on the Warburg effect (glycolytic metabolism), concretely, on the expression of genes involved in the cell reactive oxygen species (ROS) balance and levels. Accordingly, in vitro studies with primary VHL-ccRCC and 786-O cells measuring ROS levels, ROS-expression of detoxifying enzymes, and the expression of p65/NF-κB targets by RT-PCR were carried out. Furthermore, histological analyses of ccRCC samples from heterotopic mouse xenografts were performed. The obtained results show that ADRB2 blockade in ccRCC cells reduces the level of oxidative stress and stabilizes the inflammatory response. Thus, these data further support the idea of targeting ADRB2 as a promising strategy for the treatment of VHL and other non-VHL tumors.

## 1. Introduction

Renal cell carcinoma (RCC) is one of the most common tumors of the urinary system, insidious and non-responsive to chemo-radiotherapy. Clear cell renal cell carcinoma (ccRCC) is the most frequent subtype of RCC (70–80%) and is characterized by strong lipid and glycogen accumulation [1]. Although in the regular population ccRCC is sporadic, in von Hippel–Lindau (VHL) disease, they are inherited in an autosomal dominant manner. VHL patients are heterozygous for *Vhl* and the somatic deletion of *Vhl* generally occurs in ccRCC. Moreover, in VHL patients, the deletion or inactivation of the wild-type allele (Knudson’s second hit) leads to tumoral development [2].

The VHL protein is an E3 ubiquitin ligase which functions to bind the transcription factors hypoxia-inducible factors 1 and 2α (HIF-1α and HIF-2α) under normoxic conditions with their subsequent proteolytic degradation [3]. In the absence of a VHL functional protein, HIF-1α and HIF-2α are not processed to the proteasome and, hence, they accumulate in the cytoplasm and translocate to the nucleus, driving the transcription of their downstream target genes, such as those involved in proliferation, angiogenesis, glycolysis, cell dedifferentiation, extracellular matrix degradation, and erythropoiesis among others [4].

VHL is an autosomal inherited rare tumor disease with an estimated annual birth incidence of 1/36,000. Clinical symptoms include multiple tumors that appear throughout the life span of the patient, among other multisystemic tumors including CNS or retinal hemangioblastomas (HBs) and, in a notable percentage (60%), VHL patients also suffer from ccRCC.

Kidneys of VHL patients usually harbor renal cysts which develop in ccRCC [5,6]. The epithelial cells lining the cysts present an early loss of VHL protein functionality and an accumulation of HIF-2α, the most prominent HIF in RCC [7].

As of now, there is no effective treatment for VHL associated tumors, and repeated surgeries remain the only way to handle the disease, with a notable decrease in patients’ quality of life [2]. Although drugs that directly inhibit targets of HIFs, such as VEGF/VEGFR or mTOR, have been assayed in clinical trials, they have shown limited success. In particular, for advanced RCC, molecular-targeted drugs, such as tyrosine kinase inhibitors (TKIs; e.g., sunitinib or pazopanib), are the first-line treatment, but resistance appears as a consequence of adaptation to the heterogeneous tumor environment [8]. Recently and for VHL-RCC, the selective HIF-2α inhibitor Belzutifan has been approved in the USA. Belzutifan was assayed in 61 VHL patients suffering from early-stage non-metastatic ccRCCs [9,10]. The data showed a limited response, with a noncomplete response observed and 28 partial responses.

A completely different approach in the angiogenesis field is represented by β2-adrenergic receptor (ADRB2) blockers, starting with the first observations of Leauté-Labrèze, who discovered propranolol as an antiangiogenic therapy in Infantile Hemangioma [11].

On in vitro and in vivo assays, our lab has demonstrated the therapeutic properties of propranolol in VHL disease, beginning with HBs primary culture cells. Propranolol decreased viability, increased apoptotic death, and lowered the protein expression levels of HIF-1α on CNS-HBs primary cultures derived from surgical pieces. Concomitantly, gene targets of HIF-1α such as *VEGF*, *EPO*, and *SOX* appeared downregulated at RNA and protein levels [12].

Moreover, in a clinical trial (EudraCT: 2014-003671-30) with seven VHL patients harboring multiple non-tractable retinal HBs, propranolol was tested with a dose of 120 mg a day for a total of 12 months. The size or number of HBs did not increase in any of the patients. In two cases, the retinal exudates present at the beginning of the trial vanished after the treatment. As a result of the published in vitro results and the clinical trial for retina HBs, propranolol was designated as orphan drug for VHL tumors by EMA in 2017 [13]. Furthermore, tumor analysis in four VHL patients bearing ccRCC who received propranolol as an off-label treatment for retinal HBs showed a stabilization in tumor growth after three years (on average) of systemic treatment [2].

Since the therapeutical effects were driven when ADRB2 and not ADRB1 were blocked, the search for a specific β2-blocker led us to demonstrate that ICI-118,551 (ICI), an almost pure β2-blocker, had the same in vitro potential therapeutic effects as propranolol in CNS-HBs primary cultures [14], VHL-ccRCC primary cells, and the 786-O ccRCC cell line [2].

The clear-cell morphology is caused by highly active lipid and glycogen synthesis and deposition, showing a unique metabolic state of cancer [15]. While these metabolic alterations support cancer progression, they also give the opportunity to explore new targets associated with the metabolic state of ccRCC for developing new therapies.

In the same line, a recent RNAseq analysis of endothelial cells (ECs) from VHL patients (*Vhl+/−*) compared to ECs from healthy donors showed the downregulation of genes involved in reactive oxygen species (ROS) detoxification and NF-κB/p65 pathways [16].

Since inflammation and ROS production are hallmarks of tumoral processes in general [17], and of RCC in particular, the present work shows how the treatment with β-blockers decreases inflammatory targets of the NF-κB pathway and increases the expression of ROS processing enzymes, leading to a decrease in the content of ROS. Thus, previous data and the data shown in this work further support the idea of targeting ADRB2 as a promising strategy for the treatment of VHL and other non-VHL tumors.

## 2. Results

### 2.1. β2-Blockers and Viability of ccRCC Cells

We have shown previously that β2-blockers selectively decrease cell viability and induce apoptosis in VHL-CNS-HBs and VHL-ccRCC primary cells, as well as in the VHL-negative RCC 786-O cell line [2,12,14]. Further research into the effects of β2-blockers, particularly propranolol and ICI, led us to test, in detail, the effect on ROS and inflammation. Both ROS production and inflammatory processes are currently associated with cancer and tumoral processes in general and with ccRCC in particular.

Initially, we tested the effect of propranolol and ICI on the viability of the four different VHL-ccRCC primary cultures and on the 786-O cell line (Figure 1). The VHL mutations and relevant clinical data of the VHL patients from which the primary carcinoma cultures were derived are shown in Table 1.

### 2.2. β-Blockers and ROS in ccRCC Cells

According to previous RNAseq results comparing VHL-ECs versus healthy ECs, ref. [16] ECs from VHL patients showed an increase in the amount of ROS due to the downregulation of enzymes in charge of ROS processing. Hence, we measured the amount of ROS from primary cultures of VHL-ccRCC and from 786-O cells before and after treatment with β-blockers.

ROS-related fluorescence quantification, using the ROS detecting probe DCF-DA solution and the ImageJ software, showed a reduction in ROS levels of the treated cells. Representative merged images (brightfield and ROS-related fluorescence) of the primary VHL-ccRCC18 and 786-O cells are shown in Figure 2 (panel A), and histograms show the ImageJ fluorescence intensity quantification of the images. Figure 2 (panel B) shows the ROS-related fluorescence intensity measurements using a Glomax multidetection system (Promega). Fluorescence signal was normalized to cell number.

Altogether, we can conclude that the treatment of ccRCC cells with the β-blockers propranolol and ICI contribute to decrease the amount of ROS detected in defective VHL ccRCCs.

Next, we wanted to explore whether the decrease in ROS detected after the treatment could be the result of the upregulation of ROS processing enzymes. This was tested by RT-qPCRs, in the following detoxifying enzymes: *glutathione peroxidase (GPx-4)*, *nucleoredoxin*, *superoxide dismutase*, and *catalase*. The results of fou different primary cultures of VHL-ccRCC are shown in Figure 3A. *GPx-4* is upregulated by β-blockers in an effective manner in all the samples but ccRCC13.

*Nucleoredoxin* is upregulated in the four primary cultures after β-blockers treatment, although the increase is not as high as in the case of *GPx-4*.

*Catalase* is upregulated in a significant way in one of the four ccRCCs, while there is a tendency to increase with ICI in other two (*p* = 0.053; *p* = 0.09). *SOD* is also upregulated in three out of the four different ccRCCs, ccRCC13 being the only exception. It is worth mentioning that the patient where the primary culture ccRCC13 was derived was under propranolol treatment (120 mg/day) prior to the surgery.

In Figure 3B, the same RT-qPCR approach was done using the 786-O cells line. In this case, the only upregulated enzyme after β-blocker treatment is the *GPx-4*. Of note, the increase in this case was 7–8-fold compared to control expression, an increase much more pronounced than in the ccRCC primary cultures. However, the other enzymes were downregulated after β-blockers treatment in a significant way. While 786-O cell line would correspond to a stage III-IV RCC, the ccRCC primary cultures from VHL patients belong to stage I-II RCC.

### 2.3. Inflammatory Markers Downregulation in ccRCC after Treatment with β-Blockers

Albiñana et al. (2020) described how the *concentration* of the nuclear p65 component of the NF-kB pathway was reduced after β-blocker treatment of 786-O cells. The expression of the p65/NF-kB pathway was studied in VHL-ccRCCs primary cultures through the expression of some of its targets. Tested targets were *IL-6*, *IL-1β*, and *TNFAIP6*. All of them showed a significant downregulation after the β-blockers treatment. Figure 4A shows the average of four different VHL-ccRCC primary cultures. When the VHL defective 786-O cell line was analyzed, similar results were obtained but in *IL-6*. An increase in IL-6 expression was observed after betablockers treatment (Figure 4). Differences may be due to the nature of 786-O cell line, representative of an advanced stage IV ccRCC compared to the ccRCC primary cultures of VHL patients.

### 2.4. In Vivo Xenografts of 786-O Cells in Untreated and β-Blockers Treated Mice

As explained in Section 4.7, a heterotopic xenograft of 786-O cells was developed [2]. Briefly, once the tumors reached 100 mm^3^ on average, mice were treated intraperitoneally daily with vehicle, propranolol, or ICI for 20 days. Mice treated with propranolol or ICI showed a significant reduction in tumor volume, as published on Albiñana et al. (2020). Then, tumor samples were collected for histology studies.

Immunohistochemistry analysis for GPx-4 show an increase in the expression of tumors treated with ICI and propranolol, in agreement with the qPCR results shown in Figure 3 (Figure 5A). In the same way, when tumors were stained with p65, a decrease of the p65 signal was apparent in propranolol and ICI treated tumors compared with vehicle treated tumors (Figure 5B). These results are also supporting the downregulation of p65 targets detected by qPCR in Figure 4.

## 3. Discussion

RCC affects over 400,000 individuals worldwide per year [18]. There are several subvariants of RCC, ccRCC being the most frequent with 70% of the cases of RCC.

In principle, an early diagnosed ccRCC leads to surgical resection or kidney ablation. However, one in every three cases will develop metastases that are difficult to treat. Metastatic ccRCC is quite lethal and is the critical point when dealing with the ccRCC therapy [19].

The physical loss by deletion or functional loss by mutation of the *Vhl* gene seems the initial step in the development of ccRCC. In the rare tumoral disease of VHL, patients inherit a germline mutation in *VHL*, following an autosomal dominant pattern [20]. These individuals often suffer a second hit in the *Vhl* gene developing multifocal, bilateral ccRCC [21,22]. On the other hand, somatic loss of VHL function is also observed in sporadic ccRCC [23].

The VHL protein has several functions [24] but the best known is the ubiquitination of HIF-1α and HIF-2α under normoxic conditions, with their subsequent proteolytic degradation [3,25]. HIF-1α and HIF-2α promote a set of transcription genes involved in angiogenesis, metabolism, and chromatin remodeling [26].

In ccRCC, VHL loss of function activates the HIF-2α transcription factor and its downstream genes. HIF-2α is involved in angiogenesis and various other processes [27]. Thus, the current therapies have been made targeting angiogenesis by TKIs or VEGF monoclonal antibodies. However, the efficacy of these drugs is limited due to acquired resistance [28]. In this field, the VEGF/VEGFR, FGF/FGFR and PDGF/PDGFR signaling crosstalks are involved in the acquisition of this drug resistance [29]. In addition, a selective HIF-2α antagonist, PT2399, was verified to be active in tumors reluctant to the sunitinib-treatment and it was better tolerated in ccRCCs [30].

Another strategy is to decrease HIF protein levels and indirectly target by downregulating its downstream genes. We have previously published that β2-blockers decrease cell viability and induce apoptosis in VHL primary HBs and ccRCC cells including the *Vhl* negative renal carcinoma cell line 786-O [2].

In the present manuscript we have explored additional benefits of β2-blockers on the putative targets of ccRCC due to their altered metabolic processes. Inflammation and ROS are characteristic of tumors in general, and, particularly, of ccRCC.

A particular feature of ccRCC is the upregulation of glycolysis and the corresponding downregulation of oxidative phosphorylation (OXPHOS), as demonstrated by proteomics analysis [31].

Compared with other tumors such as those found in the lungs and brain, primary ccRCC tumors show a higher enrichment of glycolytic intermediates, and by contrast, low tricarboxylic acids cycle intermediates (TCA), consistent with the so-called Warburg effect [32]. The tumor microenvironment is consequently forced to reprogram energy metabolism.

In *Vhl*-deficient renal cells, due to the hypoxia triggered metabolism, several genes are found to be abnormally expressed, especially enzymes of the glycolytic pathway such as the glucose transporter 1 (GLUT1), hexokinase II (HK2), and lactate dehydrogenase LDHA [33]. Because of the increased tumorigenic metabolic activity to maintain their hyperproliferation, cells generate increased quantities of ROS resulting from upregulated glycolysis via the mitochondria and endoplasmic reticulum [34], leading to increased cell pathogenesis. Moderate ROS levels exert proliferative- and pro-survival signaling, while high ROS quantities induce cell death of tumoral cells and the microenvironment cells. Therefore, it is important to control the aberrant metabolism, redox regulation, and mitochondrial adaptions when searching ccRCC therapies [35]. In this sense, ADRB2 blockers may control the ROS levels by stimulating enzymes processing them and represent a new opportunity in cancer therapies.

We have recently demonstrated that ECs from VHL patients in heterozygote condition *Vhl* (+/−) show an increase in ROS due to the downregulation of enzymes in charge of ROS processing compared with control cells [16]. In this manuscript, we show (Figure 2) that the amount of ROS species quantified in VHL-ccRCC primary cultures and 786-O decreased in a significant manner following β2-blockers treatment (propranolol and ICI, respectively). ADRB2 blockers would be contributing to the decrease of ROS by increasing the detoxifying enzymes necessary to process ROS in ccRCC. For this reason, the expression of four different main enzymes involved in the metabolism of ROS such as GPx-4, catalase, superoxide dismutase, and nucleoredoxin were tested in four different primary cell cultures of VHL ccRCC patients and in the ccRCC cell line 786-O (Figure 3). In primary ccRCC, the RNA levels of these enzymes were increased after ADRB2 blockers treatment. However, the results in 786-O cell line were different, due to only GPx-4 showing a very relevant and highly significative increase. We should consider that the 786-O line would correspond to a grade III-IV RCC, while the ccRCC primary cultures from VHL patients are of a lower grade (grade I or II) attending to the clinical follow-up of these patients [36]. Thus, the increase on GPx4 is consistent as a result of the β-blockade in lower and higher grade ccRCC. This enzyme has been claimed as a landmark for ccRCC putative target in cancer therapy [15]. The GPx4 was discovered in 1982 as a cytosolic “peroxidation inhibiting protein” [37] and today represents a specific target for new pharmacological treatments aiming at activating cell death in cancer [38]. Remarkably, GPx4 increases following ADRB2 blockers treatment in cultured cells and in vivo, in tumor xenografts of 786-O cells from mice treated with propranolol or ICI compared with vehicle treatment.

On the other hand, some previous results [39] have demonstrated that loss of VHL was associated with an upregulation of NF-κB activity through HIF dependent induction. In addition to the HIF-transcription regulation, VHL protein was also shown to promote the inhibition of the NF-kB activation in RCC independently of HIF [40]. Peri and colleagues demonstrated the upregulation of NFκB and IFN signaling pathways in the absence of VHL [41]. In line with the last paper, we recently observed [2] by confocal microscopy that treatment with ADRB2 blockers of 786-O cells decreased the intensity of the signal of p65 in the nucleus. In the present work, we have explored in more detail the consequences of ADRB2 blockade on primary cultures of VHL-ccRCC patients. In fact, the treatment decreases the expression of NFκB/p65 targets as *IL-1**β*, *IL-6*, *CCL20*, and *TNFAIP6*. Furthermore, in in vivo xenografts of 786-O cells from mice treated with propranolol or ICI, p65 staining decreased compared with vehicle treated mice.

As discussed before, proinflammatory and Warburg effect profiles enhance malignancy of renal carcinoma, meaning a tumor surrounding invasion and metastases; the former leads to partial or total kidney resection, and the latter initiates the countdown. The ADRB2 blockers propranolol and ICI-118,551, as part of novel therapeutic strategies, have been shown to decrease the ROS content in ccRCC cells and act as anti-inflammatory drugs.

The amount of ROS is decreased by the increase of the expression of genes encoding for ROS detoxification. A reduction in ROS levels helps to maintain vascular homeostasis and to improve the renal function of the tumor surrounding tissues. Furthermore, the decrease of expression of inflammatory genes such as IL-β, IL-6, and TNF-α could provide protection from the injury caused by the carcinoma associated inflammatory response.

Previously, ADBR2 blockers had shown in vitro inhibition of carcinoma cell proliferation and antiangiogenic functions activity [2]. The additional properties found in this work, as antioxidants, by increasing the activity of detoxification enzymes, and as anti-inflammatory agents by downregulating the NFκB/p65 pathway, further supports the therapeutic value of β2-blockers for the treatment of RC, among other solid tumors.

In this sense, ADBR2 blockage shows promising therapeutic benefits in clinical use since it might delay or impair tumor malignancy and contribute to combined therapies, as has been seen in the last few years.

### Limitations of This Study

Studies on rare diseases tend to lack a large (or large enough) number of patients and consequently, it becomes difficult to get many samples.

The use of primary cultures obtained directly from surgical specimens helps to get representative samples of ccRCC. However, as primary cultures, the number of passages allowed is limited and, consequently, so is the number of experiments performed with them. Therefore, every sample is highly valuable, and we try to obtain as much information as possible from them.

Research on rare diseases must strike a balance between two opposing forces: (i) the scarcity and limitations of the samples with which this research is developed and (ii) the urgency of data generation and publication to be shared with the scientific community and patients in order for them to benefit from the knowledge. Thus, the researcher must let his imagination run wild in order to get the most out of a few samples in the shortest possible time. For this reason, results from previous publications have been included to give robustness to the data obtained, being one of the few licenses allowed in research on rare diseases.

## 4. Materials and Methods

### 4.1. RRC Primary Cell Cultures Isolation and Subcultivation

Primary cultures of *Vhl−/−* ccRCC cells were obtained from surplus tumor specimens from surgeries of VHL patients. Written informed consent was obtained from all patients for the study of their samples. All procedures had been previously approved by the Ethics Committee in accordance with general accepted guidelines for human samples (075/2017).

Briefly, the collected fresh tissue from the excess of a resected RCC was placed in sterilized tubes containing ice-cold RPMI (GIBCO, Grand Island, NY, USA). Several PBS washing steps were done before specimens were chopped into ~1 mm^3^ pieces. The resulting tissue pieces were subjected to enzymatic digestion with [1 mg/mL] collagenase I and dispase II (GIBCO) in RPMI at 37 °C. After 45 min incubation, a gentle pipetting was performed for a complete disaggregation as well as a trypsin (GIBCO) digestion at 37 °C for 30 min. Then, minced samples were centrifuged and cell pellets were suspended in RPMI with 20% FBS, 2 mM L-glutamine, and 100 U/mL penicillin/streptomycin (all from GIBCO) and incubated at 37 °C in humidity conditions. Sub-cultivation procedures followed a regular complete medium replacement (every 72 h or until the cultures were confluent).

### 4.2. RCC Commercial Cell Line

The human *Vhl−/−* epithelial RCC cell line 786-O (CRL-1932) was obtained from the American Type Culture Collection (ATCC, Manassas, VA, USA). The 786-O cells were cultured in RPMI with 20% FBS, 2 mM L-glutamine, and 100 U/mL penicillin/streptomycin (all from GIBCO) at 37 °C with 5% CO_2_ and humidity conditions.

### 4.3. ADBR2-Antagonists Treatments

Both the ccRCC primary tumors and the 786-O cell line were tested using different doses of the ADRB2 antagonists, propranolol or ICI-118,551 (Sigma-Aldrich, St. Louis, MO, USA). The lyophilized samples of propranolol and ICI were reconstituted and stored in distilled water and, for the cellular and molecular biology assays described below, were diluted in complete growth medium (described above).

### 4.4. Real-Time RT-PCR (qPCR)

The NucleoSpin RNA kit (Macherey-Nagel, Düren, Germany) was selected to isolate total RNA from the above-mentioned cells. Total RNA (1 µg/20 μL) was transcribed using the cDNA synthesis kit High-Capacity cDNA Reverse Transcription Kit supplemented with a RNase Inhibitor (both from Thermo Fisher Scientific, Vilnius, Lithuania).

To perform RT-PCR, SYBR Green PCR system (Roche, Mannheim, Germany) and iQ5 system and software (BioRad, Hercules, CA, USA) were used. The 18S mRNA, housekeeping gene, and all samples were evaluated in triplicate for the genes (and their primers) listed in Table 2.

### 4.5. Cell Viability Assay

The effect of the ADBR2-blockade on the cell viability of primary ccRCC tumors and 786-O cells was performed by analyzing the number of viable cells in the culture, based on the quantification of the presence of ATP, hence indicating metabolically active cells.

First, 5 × 10^3^ cells/well were seeded in a 96-well plate and incubated in complete medium with [0- 50- and 100-μM] propranolol or ICI for 72 h. Then, 100 μL/well of the Luminescent Cell Viability Assay kit (Promega, Madison, WI, USA) was added and gently mixed for 15 min at RT. Luminescence was measured using a GLOMAX multidetection apparatus (Promega).

### 4.6. Molecular Biology Assay: Reactive Oxygen Species (ROS) Determination

To measure the effect of the ADBR2-blockade on ROS production in 786-O and ccRCC primary cultures, the DCF-DA solution reagent was used (ROS0300, OzBiosciences, San Diego, CA, USA).

Briefly, 5 × 10^3^ cells/well were seeded in a 96-well plate and incubated in complete medium with [0–50 μM for 786-O and 0–100 μM for ccRCC] propranolol or ICI. After 72 h, cells were washed with PBS and then incubated with 100 μL DCF-DA for 30 min at 37 °C in darkness. After DCF-DA removal, 100 μL/well PBS was added and fluorescence (exc: 485 nm/em: 535 nm) was measured using a GLOMAX multi-detection system (Promega). Fluorescence signal was normalized to cell number.

Bright field and fluorescence microscopy images from the same samples were taken using a Pantera microscope and the Motic Images Plus 3.0 software (Motic, Wetzlar, Germany). FIJI-Image J software tool (NIH, Bethesda, MD, USA) was used to process and quantify the fluorescence intensities.

### 4.7. Immunohistochemistry

Protein levels of Glutathione peroxidase and the protein p65 of the NF-κB/p65 expressed in the tumors grown from ccRCC xenografts in animals treated with vehicle, propranolol, or ICI, were evaluated in paraffin-embedded tumor sections.

As published on Albiñana et al. (2020), approximately 1 × 10^6^ cells were subcutaneously injected in the left flank of 7–8-week-old NSG™ male mice (own breeding). Once the tumor volume reached 100 mm^3^ on average, the treatment of mice was initiated as follows: a daily intraperitoneal inoculation was given with either 10 mg of propranolol or ICI-118,551 per Kg of body weight. Vehicle was used as a control. The treatment proceeded for 20 days. Then, tumors were resected and paraffin-embedded for further analysis [2].

The immunohistochemical staining of the above-mentioned samples was done in 5 μm deparaffined and re-hydrated sections. Antigens were retrieved by Proteinase K according to IHC Dako protocol. They were previously blocked using Dako’s blocking solution and then incubated with an anti-p65 rabbit polyclonal antibody (sc-109, Santa Cruz Biotechnology, Dallas TX, USA) diluted 1:250 in blocking solution. The primary antibody was incubated o/n at 4 °C in a humid chamber. Next, samples were washed in PBS and incubated with HRP-conjugated antibody (K4063, Dako, CA, USA). HRP activity was amplified with Dako-EnVision + Dual Link System-HRP (Dako). Development of the immunological reaction was performed with a DAB substrate Kit (Dako) for up to 5 min. Samples were counterstained with hematoxylin 0.02% and mounted with DPX medium (Sigma-Aldrich). Images were taken using a Pantera microscope and the Motic Images Plus 3.0 software (Motic). The Vehicle group was tested as the positive control and negative controls were performed in parallel without incubation with the primary antibody. For DAB staining quantification studies, FIJI-Image J software (NIH) was used to process and quantify the fluorescence intensities. Images from 6 samples per condition were analyzed.

### 4.8. Statistical Analysis

Quantitative results are given as mean ± SD. Comparison between means was performed using Student’s *t*-test analysis. Statistical significance was defined when *p* was at least less than 0.05. The degree of significance is shown in figures as follows: (* *p* < 0.05; ** *p* < 0.01, *** *p* < 0.001).

## 5. Conclusions

VHL-ccRCC, as the second cause of death in VHL disease, urgently needs an effective systemic treatment. We have recently shown the therapeutic benefits of ADRB2-blockade in vitro, in vivo, and in an off-label treatment. Proinflammatory and metabolic (Warburg effect) profiles enhance malignancy and must be reverted. Propranolol and ICI-118,551 (ADRB2 antagonists) have shown to reduce the oxidative stress, stabilize the inflammatory response, and diminish the glycolytic pathway in ccRCC cells from VHL patients as well in a cell line. We propose ADRB2 blockade as a strategy for the treatment of VHL and other non-VHL tumors.

## Figures and Tables

**Figure 1 ijms-23-01325-f001:**
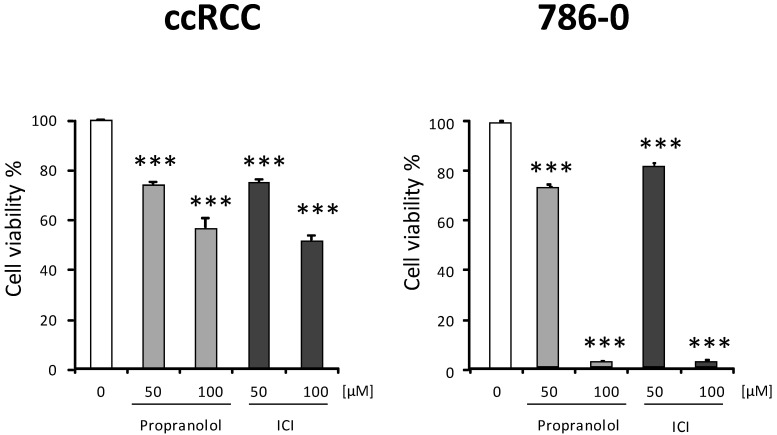
Representative cell viability of four primary VHL ccRCCs, and of 786-O cells after [0, 50, or 100 µM] propranolol or ICI-118,551 treatment for 48 h. In Figure 1, as previously reported [2], the ADRB2 antagonists propranolol and ICI dose-dependently decrease cell viability. Here, only two relevant doses of the ADRB2 antagonists (50 and 100 μM) were used in accordance with our previous experience [2]. Primary cultures from VHL patients proved more resistant to ADRB2 blockade than the 786-O cell line. Therefore, to obtain an equal number of cells, throughout the following experiments the VHL-ccRCC primary cultures of VHL-ccRCC were incubated with 100 μM of both β2-blockers, while 786-O was treated with only 50 μM. All data are based on three independent experiments for each of the four primary ccRCCs and for the 786-O cells. Error bars denote ± SEM. Student’s *t*-test: *** *p* < 0.001.

**Figure 2 ijms-23-01325-f002:**
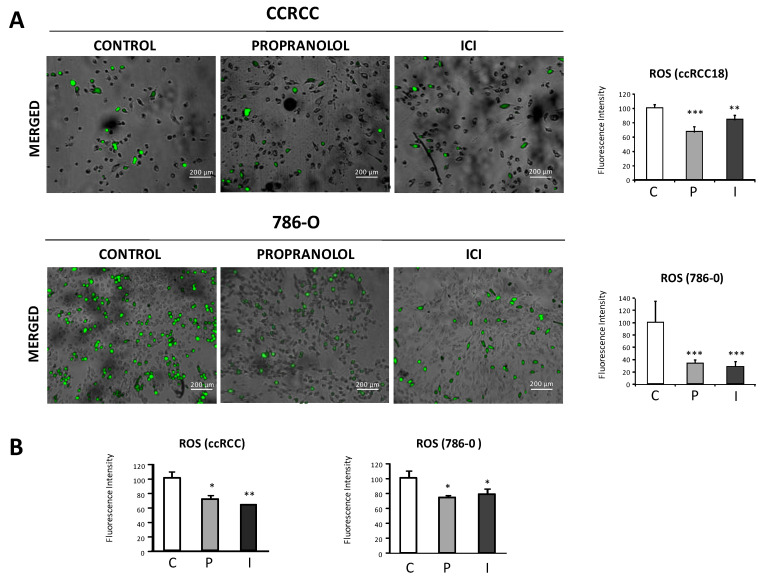
Effect of the β-blockers propranolol and ICI-118,551 on ROS levels in primary cultures of VHL-ccRCC and 786-O cells: (**A**) Brightfield and fluorescence microscopy merged images of VHL-ccRCC and 786-O cells showing the ROS expression after [0–50 μM for 786-O and 0–100 μM for ccRCC] propranolol or ICI-118,551 treatment for 72 h. Quantification of fluorescence intensities (485/535 nm) normalized to total cellular number. For quantification procedures, 12 different optic fields were taken from four replicates per condition; (**B**) The decrease in ROS species was also measured quantitatively by fluorescence, as explained in Material and Methods section. All data are based on three independent experiments. Scale bars represent 200 µm. Error bars denote ± SEM. Student’s *t*-test: * *p* < 0.05; ** *p* < 0.01; *** *p* < 0.001. C: Control, P: propranolol and I: ICI-118,551.

**Figure 3 ijms-23-01325-f003:**
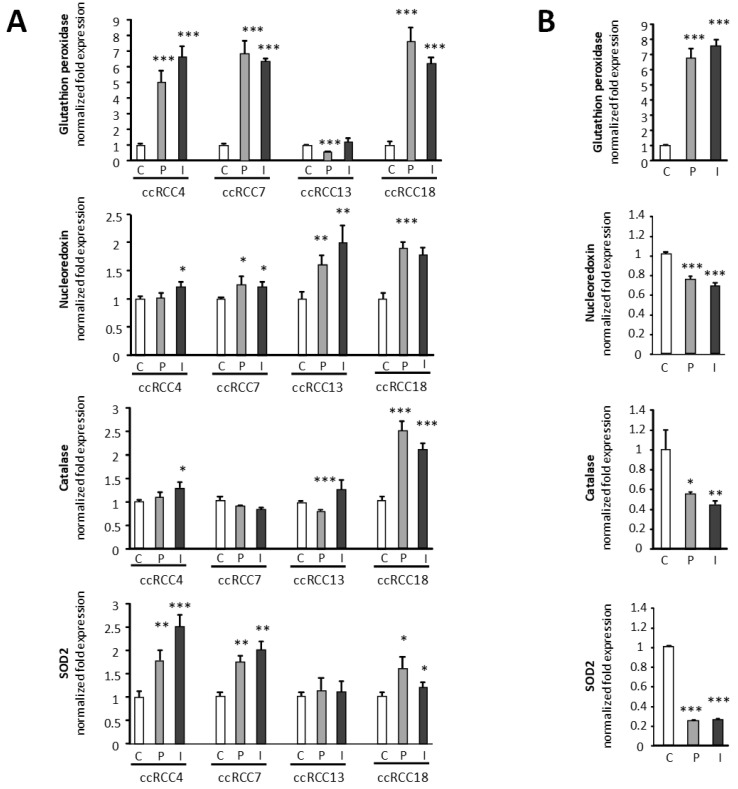
Modulation after treatment with [0–50 µM for 786-O and 0–100 µM for ccRCC] propranolol or ICI-118,551 of the mRNA expression levels of different ROS processing enzymes: *Glutathione peroxidase*, *Nucleoredoxin*, *Catalase*, and *SOD2* analyzed by RT-qPCR: (**A**) On four different VHL-ccRCC primary cultures. (**B**). On the 786-O line. All data are based on 3 independent experiments. Error bars denote ± SEM. Student’s *t*-test: * *p* < 0.05; ** *p* < 0.01; *** *p* < 0.001. C: Control, P: propranolol and I: ICI-118,551.

**Figure 4 ijms-23-01325-f004:**
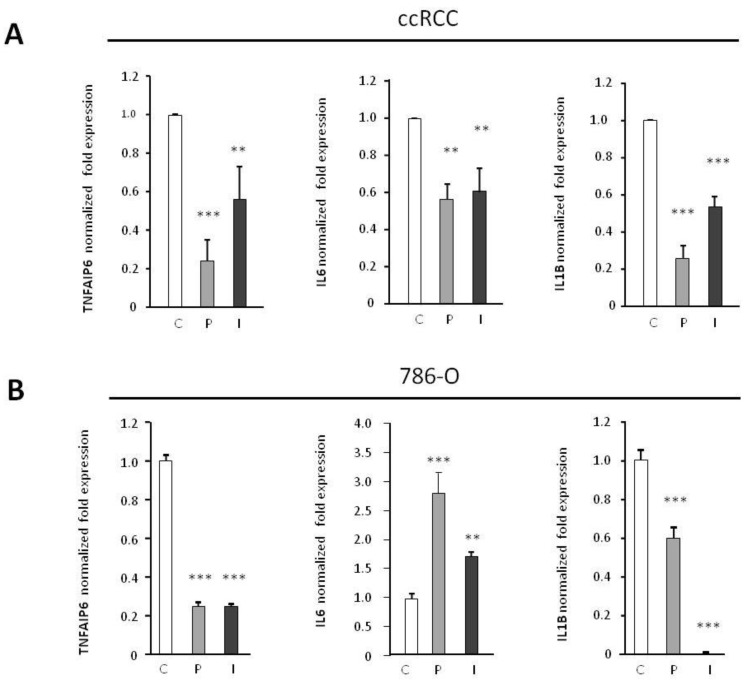
Effect of the β-blockers propranolol and ICI-118,551 on the mRNA expression levels of proinflammatory cytokines: *TNFAIP6*, *IL6*, and *IL1B*, analyzed by RT-qPCR after treatment with [0–50 µM for 786-O and 0–100 µM for ccRCC] propranolol or ICI-118,551: (**A**) Average of the four primary VHL-ccRCC cultures analyzed. (**B**) On the 786-O line. To note: in oder to increase the amount of primary ccRCC cultures from the rare disease VHL and thus, to reinforce the pattern of dose response to ADBR blockers, we have included data from tumors ccRCC4 and 7, already published [2]. All data are based on three independent experiments. Error bars denote ± SEM. Student’s *t*-test: ** *p* < 0.01; *** *p* < 0.001. C: Control, P: propranolol and I: ICI-118,551.

**Figure 5 ijms-23-01325-f005:**
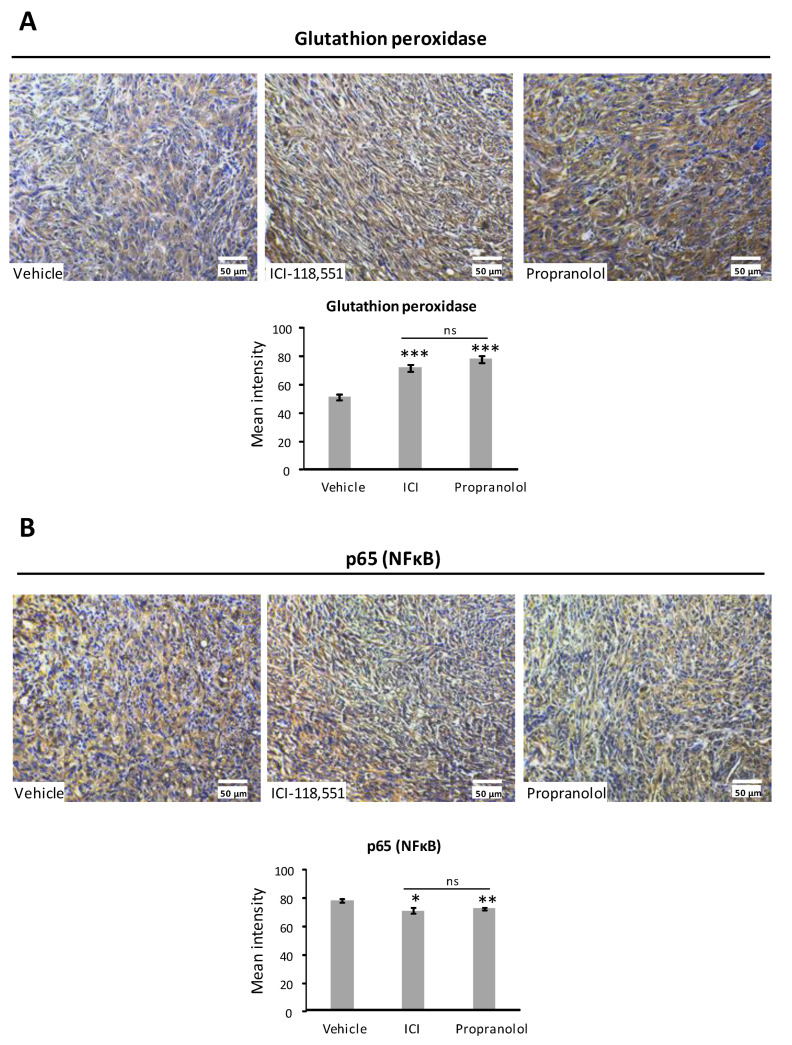
Immunohistochemical detection and quantification of Glutathione peroxidase (**A**) and p65 (NFκB) (**B**) in the 786-O tumor xenografts treated with vehicle, propranolol, or ICI. Cell nuclei were counterstained with hematoxylin. Scale bars represent 50 µm. DAB staining quantification. Hystograms show the significant increase of cellular Glutathione peroxidase on tumors treated with propranolol or ICI and the significant reduction of p65 on the same samples. All data are based on six different images per condition. Error bars denote ± SEM. Student’s *t*-test: * *p* < 0.05; ** *p* < 0.01, *** *p* < 0.001. C: Control, P: propranolol and I: ICI-118,551.

**Table 1 ijms-23-01325-t001:** Genotypes and Phenotypes of the VHL Samples.

Patient	Gender	Age	Age of Diagnosis	Location	c.DNA	Protein Change	Clinical Symptoms	Specific Treatment
ccRCC4	male	48	24	exon 3	c.501C>T	p.Arg167Trp	billateral ccRCC CNS-HB Pheochromocytoma	No
ccRCC7	male	56	28	exon 2	c.406T>A	p.Leu135X	ccRCC	No
ccRCC13	male	30	20	exon 3	c.486C>G	p.Cys162Trp	ccRCC brain trunk-and retinal-HBs	Propranolol (2 years before ccRCC surgery)
ccRCC18	female	48	30	exon 2	c.452T>G	p.Ile151Ser	billateral ccRCC medula-and cerebellum-HBs	No

**Table 2 ijms-23-01325-t002:** Primers used for q-PCR assays.

Gene	Fwd 5′–3′	Rev 5′–3′
*18S*	CTCAACACGGGAAACCTCAC	CGCTCCACCAACTAAGAACG
*Glutathione peroxidase*	TGGTGGVVTGTGTCTGTAGT	TCAGGATCTCCTCATTCTGACA
*SOD2*	AACACCTCCCTACGCCAAC	TCCCTCGTGCTTGGATTG
*Catalase*	CTCCGGAACAACAGCCTTC	ATAGAATGCCCGCACCTG
*Nucleoredoxin*	AGAAGCCTCGCCTTTTCCTA	CCTCCACCCTCACTCCATC
*IL-1β*	CTGTCCTGCGTGTTGAAAGA	TTGGGTAATTTTTGGGATCTACA
*IL-6*	CAGGAGCCCAGCTATGAACT	GAAGGCAGCAGGCAACAC
*TNFAIP6*	GGCCATCTCGCAACTTACA	GCAGCACAGACATGAAATCC

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
