# Peer review of "Blockade of β2-Adrenergic Receptor Reduces Inflammation and Oxidative Stress in Clear Cell Renal Cell Carcinoma"

_ijms, 2022, doi:10.3390/ijms23031325_

Round 1

Reviewer 1 Report

ADRB2 blockade in ccRCC cells reduces the levels of oxidative stress and stabilizes the inflammatory response. Thus, these data further support the idea of targeting ADRB2 as a promising strategy for the treatment of VHL and other non-VHL tumors. This study is well organized. Results from this study are sound. However, there are have several issues that need to be addressed. A revision is suggested.

  1. Please briefly address the methods in this abstract.
  2. Please show the unit and standard of scale bar in every photo.
  3. p65 (NF-κB) should be investigated by the nuclear isolation method. Generally, p65 is located in the cytosol fraction. Once activation, it will be translocated into a nuclear fraction.
  4. Line 360, please delete “ Cellular Biology Assay”
  5. Please discuss the clinical implications of this study.
  6. Please discuss the limitation of this study.

Author Response

Dear Editor and Reviewers,

We thank the editor and the two reviewers for the time they spent looking over the manuscript. We also appreciate the comments and suggestions done to improve quality and the clarity of the manuscript.

As an attached word file is our response to each point raised by the reviewers.

All major changes made to the original manuscript have been highlighted in red. Furthermore, the English of the text has been edited by a native speaker to improve the quality of it.

We hope that we satisfyingly addressed them all and that the manuscript will be now suited for publication.

Sincerely,

Luisa M Botella and Angel M Cuesta.

Reviewer 2 Report

In this study, Albinana et al build on previous work from their lab and assess the effect of two ADRB2 blockers -propanolol and ICI-118551, which were previously shown to reduce tumor growth in mouse xenografts of the 786-O VHL-renal cell carcinoma cell line-, in the expression of inflammation-related genes, ROS processing enzymes and production of ROS by patient derived VHL-ccRCC cells and the 786-O cell line. Consistent with previously published results (ref. 2, Albinana et al, 2020), authors found reduced viability of ccRCC/786-O cells treated with ADRB2 blockers and changes in gene expression of inflammation-related genes. Further, they also reported a decrease in the production of ROS and changes in mRNA expression levels of ROS processing enzymes upon treatment. Since accumulation of HIF-1a and HIF-1b is a hallmark of VHL pathogenesis, authors reasoned that their results support targeting ADRB2 as a therapeutic approach against VHL tumors.

Overall, the experimental design is correct, although data presentation must be improved for clarity and transparency. Conclusions are only partially supported by the data and results and novelty of this study should be discussed in more detail, i.e. with clear explanations or proposed hypotheses that would justify the discrepancies between the experimental results and proposed conclusions.

Specifically:

  • Table1 should include all relevant information and potential confounding factors for the cohort under analysis (i.e. age, sex, age at diagnosis, treatment, etc). Authors only mention that ccRCC13 was under propranolol treatment prior to surgery, but this information is missing for the rest of the samples analyzed in this study.
  • Data on Figure 1 has been already partially published (see fig 1B,C of Albinana et al, 2020, where viability of ccRCC4, cRCC7 and 786-O was assessed); please justify why it is necessary to include it here again and make it clear and explicit that some of the data included in Figure 1 comes from a previous publication.
  • Figures 1-4 should include all individual data points, n=3 allows for easy visualization of the full dataset, there is no reason to not to represent individual values together with the information already present in the barplots.
  • In Figure 2B, was cell number considered for the quantifications? (i.e. is fluorescence signal normalized to cell number?). This is not indicated neither in the figure legend nor in the methods section, please specify.
  • 786-O cells seem to be more sensitive to ARDB2 blockers than primary ccRCCs, but they downregulate 3 out of the 4 measured ROS processing enzymes, how do the authors explain this? Can the observed effect in cell viability and ROS production be exclusively attributed to the increase in glutathion peroxidase? Has this been formally tested (i.e. phenotype-rescue experiments). If not, what alternative mechanisms could be involved in the observed phenotype?
  • In Figure 4A: how long were the cells treated with the drugs? Data for 3/4 ccRCC samples was already published in Fig. 4B of Albinana et al, 2020. Please, justify why it’s necessary to include it again in this manuscript and clearly indicate in the text that most of the data in this figure comes from a previous publication. If the experimental setup is different here than in your previous manuscript, please explain rationale behind the experiment.
  • Figure 4: how do authors interpret the increase in IL6 in 786-O cells? Do authors think that IL-6 expression is irrelevant for the effects induced by ADRB2 blockers? A more high-throughput analysis of gene expression (microarray/RNA-Seq) or protein abundance (mass spectrometry) of ccRCC or 786-O cells (optimally in an in vivo setup) would aid understanding the effects of ADRB2 blockers both in terms of inflammation and ROS production.
  • Figure 5 should include quantifications of the immunohistochemistry (IHC) stainings presented, images are not enough to support authors’ conclusions. Is there a reason why none of the other inflammation-related proteins / ROS processing enzymes assayed by qPCR were also tested by IHC in this xenografted tumors?
  • In figure 5, it is not clear whether the tumors were generated de novo for this study or if they come from ref. 2, Albinana et al., 2020 (as it seems to be suggested in lines 214-218), please clarify it in the text (methods section would suffice).
  • Would it be technically possible to generate xenografts from any of the patient-derived ccRCC cells included in this study? If so, analyzing tumor growth, expression of NFKB targets / inflammation-related / ROS-related genes (optimally at large scale, to avoid gene selection biases) and ROS production upon ADRB2 blocking in this pre-clinical model would be much more relevant to understand the response mechanisms triggered by ARDB2 blockers and translate findings to the clinical setting.

Minor points:

- Please avoid using acronyms in the paper title (ccRCC -> clear cell renal cell carcinoma; ADRB2 -> B2-adrenergic receptor) as this makes it more difficult for non-expert audiences to search for relevant literature.

- Figure 1: Y-axis scale should not exceed 100 (max. viability), please adjust accordingly.

- Line 39, please correct English: “shown” -> showed / have shown.

- Line 115, please correct English: “decrease” -> decreases.

- Please, rephrase lines 133 to 139 for clarity and correct English.

- Lines 152-156 should be included as figure legend, not main text.

- Line 201: it is not clear to what “average” the text refers to, please clarify.

- Line 214: there is no section 2.5 in the paper, please correct.

- Line 373: please correct exponential number format.

Author Response

Rebuttal letter (manuscript #ijms-1544530, Blockade of β2-adrenergic receptor reduces inflammation and oxidative stress in clear cell renal cell carcinoma)

Dear Editor and Reviewers,

We thank the editor and the two reviewers for the time they spent looking over the manuscript. We also appreciate the comments and suggestions done to improve quality and the clarity of the manuscript.

As an attached word file is our response to each point raised by the reviewers.

All major changes made to the original manuscript have been highlighted in red. Furthermore, the English of the text has been edited by a native speaker to improve the quality of it.

We hope that we satisfyingly addressed them all and that the manuscript will be now suited for publication.

Sincerely,

Luisa M Botella and Angel M Cuesta.

Round 2

Reviewer 1 Report

My questions had been well addressed. This submission is acceptable at this present vision. 

Reviewer 2 Report

Authors have very carefully answered all the points I raised in my initial revision and have updated the manuscript accordingly. Data presentation is now much clearer, experimental methods easier to follow and the rationale behind the inclusion of pre-published data is appropriately presented and included in the revised manuscript. Authors have also included a section discussing the limitations of the study, which I found very relevant and pertinent, and which overall provides a better view on the hurdles associated to doing research on rare diseases. The discussion of the experimental results has also been expanded. Together, all these updates have significantly improved the quality of the manuscript and made it worthy to be considered for publication. However, there are a few points that require further revision:

  • Regarding the representation of all individual data points in figures 1-4: it is perfectly fine to just include one representative experiment out of 3 if all of them showed the same result. In this case, please include all individual points from the technical replicates within the representative experiment which is being represented. Alternatively, individual points could represent the mean of the 3 technical replicates for each biological replicate instead. Both representations provide a good view of either technical variability (option 1) or biological variability (option 2).

  • Figure 4: while I understand the reason behind the original representation of the data, I believe the new proposed representation is more informative, so I would suggest to include it instead of the original. I agree with the authors that specifically mentioning the lack of expression of IL1B in ccRCC18 would be necessary.

  • This is not strictly necessary, but authors may want to include (totally or partially) their discussion/speculation about the different responses observed in 786-O vs primary ccRCC-VHL cells, as I found it explicative and informative.

  • Regarding Figure 5: please include individual data points for the 6 images analyzed per condition.

Finally, I would like to thank the authors for their thorough assessment of all points discussed in the initial revision of the manuscript.